# Predicting Final User Satisfaction Using Momentary UX Data and Machine Learning Techniques

**Kitti Koonsanit *** and **Nobuyuki Nishiuchi**

Department of Computer Science, Graduate School of Systems Design, Tokyo Metropolitan University, Tokyo 191-0065, Japan; nnishiuc@tmu.ac.jp
* Correspondence: koonsanit-kitti@ed.tmu.ac.jp

**Abstract:** User experience (UX) evaluation investigates how people feel about using products or services and is considered an important factor in the design process. However, there is no comprehensive UX evaluation method for time-continuous situations during the use of products or services. Because user experience changes over time, it is difficult to discern the relationship between momentary UX and episodic or cumulative UX, which is related to final user satisfaction. This research aimed to predict final user satisfaction by using momentary UX data and machine learning techniques. The participants were 50 and 25 university students who were asked to evaluate a service (Experiment I) or a product (Experiment II), respectively, during usage by answering a satisfaction survey. Responses were used to draw a customized UX curve. Participants were also asked to complete a final satisfaction questionnaire about the product or service. Momentary UX data and participant satisfaction scores were used to build machine learning models, and the experimental results were compared with those obtained using seven built machine learning models. This study shows that participants' momentary UX can be understood using a support vector machine (SVM) with a polynomial kernel and that momentary UX can be used to make more accurate predictions about final user satisfaction regarding product and service usage.

**Keywords:** user experience; UX; UX evaluation; satisfaction; prediction; machine learning

## 1. Introduction

User experience (UX) refers to all aspects of how people interact with a product or service. UX emphasizes the experiential, affective, meaningful, and value aspects of human–computer interaction and product ownership but also includes a person's perceptions of practical aspects such as utility, ease of use, and product or service efficiency. UX is highly context-dependent, subjective, and dynamic [1], as it concerns an individual's performance, feelings, and thoughts about the product or service. Moreover, these can change over time as circumstances change.

The design processes of products and services are often evaluated using the comprehensive full user experience (UX) evaluation method for time-continuous situations [2,3]. From the first to the final stage of usage, the user's emotions and perceptions can change continuously through the receipt of multiple stimulatory experiences while using products or services.

After usage, users are asked about their overall satisfaction as an indicator of final user satisfaction regarding one or more aspects of the product or service. Answers relating to final user satisfaction are often expressed on a scale that includes negative and positive values, ranging from −10 to +10 [4,5], with higher scores indicating higher satisfaction.

In particular, final user satisfaction following their experiences has been considered an extremely important factor in users' decisions about further use or recommending the products or services to other people [6]. However, final user satisfaction reported after use may be imprecise because it varies according to situations such as user activities, as shown in Figure 1.

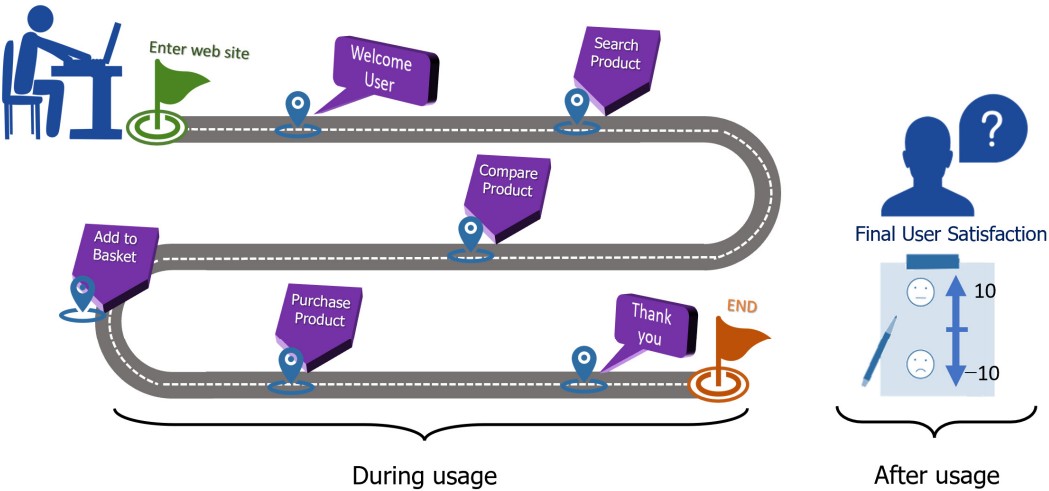

**Figure 1.** Evaluation of final user satisfaction after website usage.

Figure 1 shows several stages of usage, each of which may impact the user's emotions and perceptions and, in turn, affect final user satisfaction. Roto et al. presented the *User Experience White Paper*, a document reporting Dagstuhl Seminar's results on categorizing user experience from the viewpoint of time axis [7]. In that document, the importance of analyzing UX across time was underlined. There are four types of UX—anticipated, momentary, episodic, and cumulative (Figure 2)—each of which is defined based on usage time: (1) anticipated UX relates to the period before first use; (2) momentary UX relates to the period during usage; this type refers to any perceived change that occurs during the interaction, at the very moment [8]; (3) episodic UX relates to the period after usage; and (4) cumulative UX relates to the entire period, including from before first use, during usage, and after usage. The four types of UX can affect the final user satisfaction.

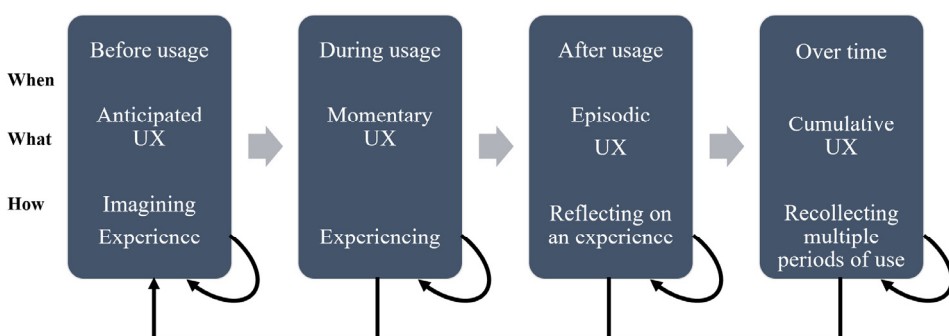

**Figure 2.** The four types of user experience (UX), adapted from ref. [7].

Previously, Kurosu et al. defined the meaning of user satisfaction as the vertical axes of satisfaction based on the UX graph being the same as user satisfaction [3,9]. Final user satisfaction means satisfaction after momentary UX. In the present study, we defined the meaning of final user satisfaction similar to Kurosu et al., that is, user satisfaction after finished usage [3,9]. Based on this definition, the final user satisfaction is similar to episodic UX and cumulative UX.

In the past decade, there have been many studies of episodic UX and cumulative UX, with most focusing on only one type [10]. Many studies have tried to estimate the satisfaction of users using various machine learning techniques [11–13]. Matsuda studied the satisfaction level of tourists during sightseeing by using the tourists' unconscious, natural actions [11]. They conducted experiments with 22 tourists in two different touristic areas in Germany and Japan. Their results confirmed the feasibility of estimating both the

emotional status and satisfaction level of tourists. Cavalcante applied machine learning techniques including decision trees, support vector machines and ensemble learning to predict customer satisfaction from service data [12]. The results indicated that the development of an intelligent algorithm may assist in identifying customer satisfaction. Kumar presented a machine learning approach to analyze tweets to improve customer experience [13]. They found that a machine learning approach can provide better classifications for customer satisfaction in the airline industry. All of the aforementioned studies gathered data and measured satisfaction using episodic UX and cumulative UX from sensors or devices. However, during actual usage by customers, there are many external factors that can affect their satisfaction.

A major problem with assessments of both episodic UX and cumulative UX, however, is that the graph or curve is recorded after the user has finished the task. Moreover, studies of both episodic UX and cumulative UX evaluation have employed the usage time to collect data, rather than using other methods to evaluate UX type. It has been pointed out that these two types of UX evaluation require participants' dedication over time [14], with assessments typically spanning intervals ranging from a few days to a month or more. Therefore, this paper focuses on momentary UX and examines the emerging role of momentary UX in the context of final user satisfaction.

Momentary UX has been measured and evaluated by questionnaire (subjective evaluation) surveys, with question items for each step of the experience. However, comprehensive evaluation of subjective answers to these questionnaires is difficult because the conventional methods of analyzing subjective evaluation may not adequately relate to the momentary UX. Instead, the quality of conventional analytic methods is determined by the experts and is directly dependent on their level of expertise. Furthermore, multiple-evaluation comparisons may be difficult due to the variety of checklists used and difficulty in quantifying expert opinions [15].

Some previous studies have measured UX at each stage of usage [16,17]. Despite the relationships between momentary UX and episodic UX, various factors will interact intricately during the actual user experience, and the final satisfaction (episodic UX) will be determined from the accumulation of experiences at each stage. This view is supported by Sánchez-Adame [18], who writes that, as an example, the user might experience a strong, albeit temporary, negative reaction when evaluating momentary UX during usage, but when episodic UX is measured again after usage, the user may be more likely to prioritize good aspects over bad ones. These data are interesting because the evaluative judgment at each stage is related to overall final satisfaction with the product.

UX is subjective, relating to an individual's feelings and satisfaction. Expert evaluations of UX may lead to bias, and such opinions are not easily quantifiable. Humans are prone to many types of bias. Despite algorithms having their own challenges, machine learning algorithms may conceivably be capable of producing more fair, efficient, and bias-free outcomes than humans. This study aimed to predict final user satisfaction by combining momentary user experience data and machine learning techniques. Our hypothesis is that machine learning will perform well on momentary user experience data in the prediction of final user satisfaction.

## 2. Literature Review

Customer relationship management (CRM) is an approach to maintain positive customer relationships and to improve customer satisfaction [19]. This new management process is aimed at improving the business and customer relationships, strategically regarding the core enterprise business customers as an important resource, meeting customer needs through the improvement of customer service and in-depth analysis of the customer, so that enterprises can maximize customer satisfaction and loyalty, establish mutual long-term stable and trusted relationships, thereby maximizing customer lifetime value [20].

Furthermore, CRM provides data and information about customers, such as their feelings, shopping behavior, and product consumer habits, among others. These user data

and information provide essential feedback from the customers' perspective, including their opinions, favors, preferences, and past experiences. The information thus obtained is used to improve communication with customers to create value and satisfaction [21]. CRM analytics can help facilitate better product or service decisions.

Some recent research reported that customer relationship capability and CRM technology in the service industry are important variables in building customer satisfaction post-purchase [22,23]. In other related work, Taufik proposed a method to utilize user data from a CRM system. They developed an online analytical processing (OLAP)-based analytical CRM system to analyze customer data and classify it into two main segments, based on geography and demographics [24]. The benefit of his approach is that the analytical process can operate upon user data from various dimensional perspectives to quickly capture customer needs in real-time. This analytical CRM system can be easily accessed by managers to make decisions. A recent study presenting an approach to processing user data and information obtained from the CRM system to satisfy customers concluded that CRM plays a major role in increasing customer satisfaction. Thus, it improves both in-depth customer knowledge and higher customer satisfaction [19].

One highly interesting aspect of customer data from CRM is user experience [25]. Several articles regarding the measurement of UX to gauge satisfaction have been published [26]. In the modern digital world there are many methods to gather UX data via automation technologies, such as interactive responses, and online questionnaires [27]. In most approaches, UX is generally measured by a questionnaire or survey method. However, evaluation of the final user satisfaction with products and services using UX questionnaires has been considered challenging because it is difficult to measure the final user satisfaction. Due to differences in user experience for each user, both humans and computers have had difficulty in classifying these data for developing or improving products and services. Furthermore, although answers from UX questionnaires can provide abundant information about a range of feelings, their high complexity substantially increases the computational burden in interpretation for experts. In this context, a new approach integrates knowledge between UX obtained from CRM and intelligent systems with machine learning techniques. This approach can be of practical value for customer relationship management by improving understanding of user satisfaction [28,29].

### 2.1. User Experience (UX)

The term "user experience" refers to a person's overall experience of interacting with a product or service [30]. UX covers not only direct interactions with the product, for example, but also how the resulting experience fits into the overall task completion process. Every interaction between the user and product or service is factored into the overall user experience. As a result, final satisfaction with respect to these UXs has been regarded as highly important in the users' decision to continue using or recommending the products or services to others [6].

### 2.2. UX Evaluation Method (UXE)

Many approaches to evaluating UX have been proposed [31], with studies proposing various methods and ways of categorizing the data. Some UXEs that might appear similar are, in fact, not. We decided to classify the many UXEs into five groups by considering "periods" of experience [31]. Methods are defined as uniquely applicable to a specific period, such as before, during, or after usage, and are sensitive to that period's characteristics, for example, momentary UXE, episodic UXE, and cumulative UXE. Accordingly, we classified the UXE methods as follows:

- Before usage (prior to interacting with products/services);
- Momentary (a snapshot, e.g., perceptions, emotions);
- Single (a single episode in which a user explores design features to address a task goal);
- Typical test session (e.g., 100 min in which a user performs a specific task).
- Long-term (e.g., interacting with products/services in everyday life).

When these five methods are applied, momentary evaluation is considered as short-term, and a reliable way to capture feelings and user experiences during usage. Although the short-term evaluation method may miss data between stages of user experience [17,32], it is one of the most reliable methods [33] because it records time-varying subjective experiences, reducing response biases and memory distortions. This is reflected in UX's dynamic nature in the longer term. During momentary use, users may experience various unexpected events during their interaction. As momentary data logs can be useful for UX evaluation, we decided to use momentary evaluation for this research.

Most research on UXs [31] has described changes in user experience over time. Examples include the UX Curve method [32], UX Graph method [3,9], and iScale method [34] as shown in Table 1.

**Table 1.** Summary of user experience evaluation through user experience (UX) curve, UX graph, and iScale.

| Approach. | Description |
| --- | --- |
| UX Curve [32] | UX Curve is a tool for drawing a timeline and a horizontal line that splits positive and negative experiences. |
| UX Graph [3,22] | UX Graph is a tool for drawing the degree of satisfaction on a time scale. It is an improved version of the conventional UX Curve. |
| iScale [34] | iScale is a tool for the backward-looking expression of long-term user experience data. |

These three UXE methods involve self-reported measurements over time, whereby the participants report their feelings and emotions in the form of line graphs drawn by hand. However, these methods are not suitable for determining final user satisfaction because drawings are made after plotting each episodic event. This means that the UX curve and UX graph are drawn, and the points specified only after the task is finished, which makes the method time-consuming [3]. Furthermore, most UXE methods are used to describe only how user experience changes during usage. Untidy handwriting means that characters in text can be difficult to read [32], so that evaluation results may be difficult to analyze and interpret. Thus, iScale, UX curve, and UX graph fall short of the requirements for appropriate final user assessment.

### 2.3. Classification Techniques

The various UX data concerning momentary usage can be problematic when it comes to analysis. For example, UX data are not simply one-dimensional, and each questionnaire may have a different scoring range. This makes for characteristically tedious work that is considered repetitive by UX researchers; in particular, the use of human labor to explain and analyze user satisfaction is not optimal. Consequently, the field of feedback and user satisfaction from pilot product studies has shown little progress or improvement over time. Hence, machine learning methods that facilitate analysis and understanding of final satisfaction have long been sought [35].

The type of machine learning algorithms used in the present study were determined by multiple factors, ranging from the type of problem at hand to the type of output desired, including type and size of the data, available computational time, number of features, and observations in the data. All such factors are important when choosing an algorithm before conducting research. Many scholars hold the view that support vector machines (SVMs) [36] can efficiently perform non-linear classification when the correct kernel and an optimal set of parameters are used [37]. Recent research has suggested that SVMs can be used for classification as well as pattern recognition purposes, especially with speech and emotion data [38]. Furthermore, algorithms such as SVM, K-nearest neighbors (KNN) [39], and logistic regression [40] are easy to implement and run [41]. By contrast, neural networks with high convergence time require significant time to train the data.

We chose seven machine learning algorithms as simple and easy-to-build classification models. We compared these seven different methods including polynomial kernel SVM, radial basis kernel SVM, linear kernel SVM, sigmoid kernel SVM, logistic regression [40], K-nearest neighbors [39], and multilayer perceptron [42] as shown in Table 2.

**Table 2.** Summary of classification techniques.

| Approaches | Description |
| --- | --- |
| Support Vector Machine with Polynomial Kernel Function | The SVM algorithm uses the best line to separate n-dimensional space into classes by the hyperplane. The learning of the hyperplane is processed by transforming the problem using Polynomial Function [40]. |
| Support Vector Machine with Radial Basis Kernel Function | SVM models classify data by optimizing a hyperplane that separates the classes using Radial Basis Kernel Function [40]. |
| Support Vector Machine with Linear Kernel Function | This classifier is formally defined by a separating line. The learning of the hyperplane is processed by transforming the problem using linear algebra [40]. |
| Support Vector Machine with Sigmoid Kernel Function | SVM models process data points by drawing decision boundaries with the Sigmoid Kernel Function [40]. |
| K-Nearest Neighbors | K-Nearest Neighbors uses the label of data points surrounding a target data point to define the class label by a plurality vote of its neighbors [39]. |
| Logistic Regression | Linear Regression is a technique to predict a continuous output value from a linear relationship. However, the output of Logistic Regression will provide a value between 0 and 1, a probability [40]. |
| Multilayer Perceptron | A multilayer perceptron (MLP) is a technique to classify the class label. It is the same structure as a single layer perceptron with one or more hidden layers. It can only classify separable cases with a binary target (1, 0) [42]. |

### 2.4. Sampling Techniques

The number of data points plays an important factor in the creating of machine learning models. The issue of limited amounts of data has received considerable critical attention. Investigators have recently examined the effects of the sample size on machine learning algorithms. Although it may be hard to determine the exact number of data points that any given algorithm requires, some studies demonstrate that using small sample sizes for building classical machine learning model leads to better performance [43]. Other studies discuss the number of samples per class for small general datasets [44].

A lack of sufficient data may lead to serious problems, such as an imbalanced distribution across classes [45]. Because many machine learning algorithms are designed to operate on the assumption of equal numbers of observations for each class, any imbalance can result in poor predictive performance, specifically for minority classes. To solve the problem of imbalance in distribution, we covered a suite of data sampling techniques to generate alternative, synthetic data [46]. The sampling method is obtained from the creation of new data or a pre-existing original dataset, and then used to create a new classification model with the machine learning method. Different sampling techniques are available for imbalanced datasets [46–51].

### 3. Methods

#### 3.1. Proposed Framework

In the UX approach, classification analytics-built models rely on momentary UX data to predict user satisfaction levels. Our proposed framework aims to predict final user

satisfaction guided by momentary UX data to answer satisfaction-related questions. The evaluation process workflow architecture is shown in Figure 3.

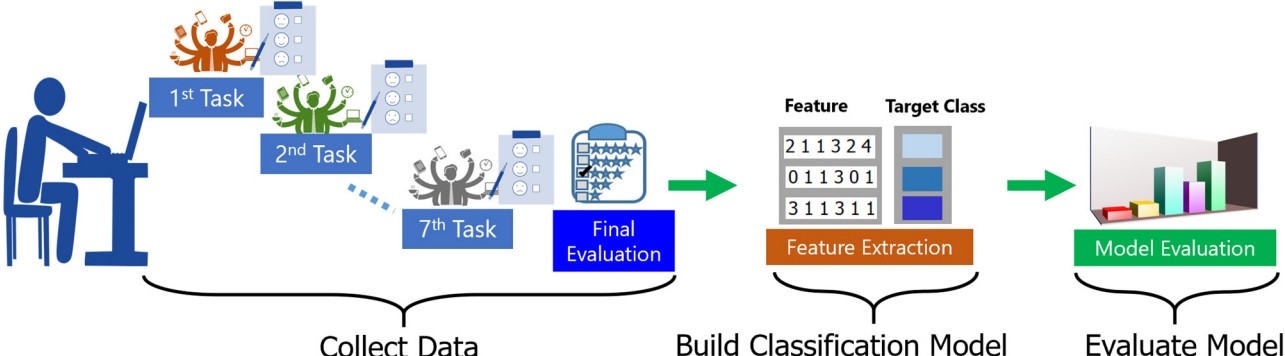

**Figure 3.** Workflow of our proposed evaluation process.

Our proposed framework was organized into three steps. First, data collection involved gathering and measuring information through satisfaction survey questions. Second, we built a machine learning process to classify the final user satisfaction into different classes. To confirm the effectiveness of the proposed framework for a product and a service, two experiments were run, using different momentary UX data. Each experiment included momentary UX data from satisfaction survey questions representing changes in emotion. Experiment I included momentary UX data from visiting a travel agency site, a website service. Experiment II included UX usage data from Google Nest Mini [52], which is a smart AI speaker product. Finally, after the classification model was built, we evaluated it using leave-one-out cross-validation and data splitting techniques.

### 3.2. Experiments

Two experiments were conducted: the first concerned use of a service in the form of a travel agency website; and the second concerned use of a product, namely Google Nest Mini.

### 3.2.1. Travel Agency Website (Service Group)

Fifty healthy university students aged 21 to 24 years were recruited as participants. The main reason for choosing people of this generation is that they typically have a better understanding of how to use products by themselves, with fewer gaps in relevant knowledge and education. We used snowball sampling to recruit participants [53]. This is a network-based sampling method that starts with a convenience sample and incentivizes participants or respondents not only to participate in the survey themselves but also to ask their contacts in the target population to participate. Snowball sampling is similar to peer-to-peer marketing, which is the best sampling method for new products or brands to reach new customers via word of mouth from one person to another [54]. For the main experiment, participants confirmed that they understood the procedure, and they responded to seven satisfaction survey questions concerning the travel agency website, as shown in Figure 4. Before they started the task, we instructed them to use the agency website to find a place they wanted to visit once in their life. All participants appeared to perform the task attentively.

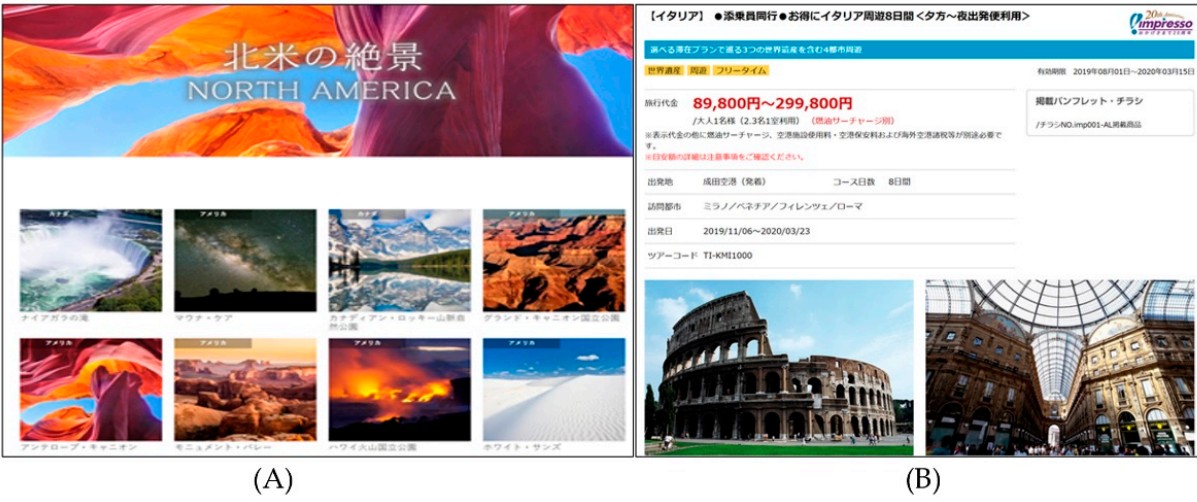

**Figure 4.** An example of a travel agency website interface (**A**) Attractive places (**B**) Tour information with a price range (https://www.his-j.com, accessed on 10 January 2020).

Ranges of rating scales can vary widely; for example, from 1–10 points to 1–100 points. One view is that overdetailed point scales may produce more variance. According to Spool (2015), enlarging a scale to see higher-resolution data may reveal that the data are meaningless [55]. Moreover, other evidence [56] suggests that a semantic differential scale may be appropriate for measuring satisfaction, with bipolar alternatives such as positive/negative, good/bad, helpful/unhelpful, and useless/valuable. These considerations led us to design and combine rating scales and the semantic differential for this evaluation. For momentary UX evaluation, we used a 21-point scale that included negative values ranging from −1 to −10 and positive values ranging from 0 and +1 to +10 [5]. Adaption of the 21-point scale was done with reference to the UX graph form [9]. A new classification model was then built using these data and the machine learning process. Finally, we measured the classification model's efficiency in terms of accuracy, precision, and recall.

Participants went through the six steps of their task in fixed order, completing the customized UX curve after each one (steps 1–6), as shown in Table 3 and the left side of Figure 5. This procedure is often implemented in actual service or product usage. Then, after completing the seventh step, they recorded "final satisfaction" based on several experiences, as shown in Figure 5. The seventh step was conducted for the study only and is not integral to the UX of the actual website itself. The data obtained in step 7 were used as a target variable for supervised learning. The right side of Figure 5 shows a final user satisfaction score of 4, based on a 21-point scale.

**Table 3.** Details of the travel agency (service) task.

| Steps | Directions |
|---|---|
| 1st | Find where you want to visit once in your life. Then, evaluate user satisfaction. |
| 2nd | Find the country of interest. Then, evaluate user satisfaction. |
| 3rd | Visit the homepage of the travel agency website. Then, evaluate user satisfaction. |
| 4th | View information on the travel agency website. Then, evaluate user satisfaction. |
| 5th | Select a tour in which you are interested. Then, evaluate user satisfaction. |
| 6th | Select and then purchase a favorite tour. Then, evaluate user satisfaction. |
| 7th | Evaluate your final user satisfaction with the travel agency website. |

**Table 4.** Original data scale was reduced by scaling down to improve the predictive performance.

| Meaning of Satisfaction Rating | Dataset I | | Meaning of Satisfaction Rating | Dataset II | |
|---|---|---|---|---|---|
| | Original Data | After Shrinking | | Original Data | After Shrinking |
| Extremely satisfied | 10 9 8 | 3 | Extremely satisfied | 10 9 8 | 2 |
| Satisfied | 7 6 5 4 | 2 | | 7 6 | |
| Slightly satisfied | 3 2 1 | 1 | Satisfied | 5 4 3 2 1 | 1 |
| Neutral | 0 | 0 | Neutral | 0 | 0 |
| Slightly unsatisfied | −1 −2 −3 | −1 | Unsatisfied | −1 −2 −3 | −1 |
| Unsatisfied | −4 −5 −6 −7 | −2 | | −4 −5 | |
| Extremely unsatisfied | −8 −9 −10 | −3 | Extremely unsatisfied | −6 −7 −8 −9 −10 | −2 |

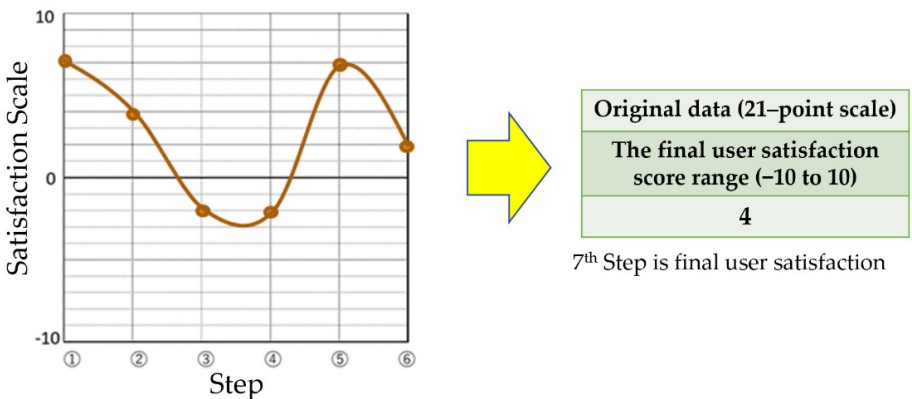

Steps 1–6 are steps/instructions.

**Figure 5.** Customized UX curve for data collection. Six satisfaction datapoints and one final satisfaction datapoint were obtained for each participant. The resulting dataset for building the model consisted of a $7 \times 50$ matrix (seven features, 50 participants). Because the original dataset revealed an accuracy score less than 0.50, we considered that results obtained using the 21-point scale were insufficiently accurate. Thus, we scaled down, converting the original dataset into two datasets based on a seven-point scale and a five-point scale (see Table 4). Dataset I comprised seven classes (from −3 to 3), and Dataset II comprised five classes (from −2 to 2). After shrinking, in actual results, we found that Dataset I comprised six classes due to zero samples in one class, while Dataset II still comprised five classes.

Furthermore, we found that the number of samples per class increased when the number of classes decreased. One advantage of shrinking is that the increased number of samples per class can be useful for building machine learning models.

Before processing the dataset, we used variance inflation factor (VIF) to check for multicollinearity of predictor variables (six answers about satisfaction score from the six-item questionnaire) where the dependent variable was final user satisfaction. VIF values exceeding 5 or 10 indicate problematic collinearity [57]. We confirmed that all our VIF values were under 5.

### 3.2.2. Google Nest Mini (Product Group)

Twenty-five university students aged 21–24 years were recruited as participants. In this experiment, the task was to remove the smart speaker (Google Nest Mini, as shown in Figure 6) from the box, set it up, and start using it through 12 steps in a fixed order. At the end of each step, the participants recorded their satisfaction on a form based on the customized UX curve. The data from the first experiment (service group) show good accuracy when rescaled in the form of a UX curve. Therefore, in this experiment, we used a new form based on the customized UX curve with a 15-point scale ranging from −7 to +7. Their final user satisfaction for the product was recorded after the experimental task was completed.

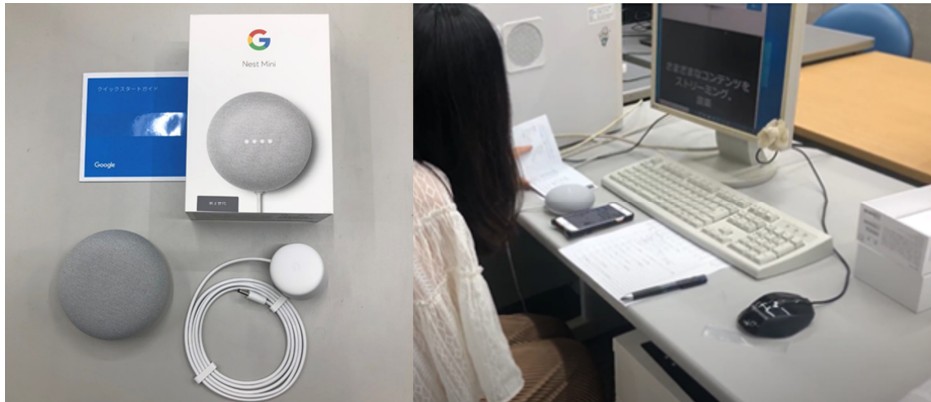

**Figure 6.** Using Google Nest Mini.

The task assumed that a new smart speaker was purchased, removed from the box, set up, and made ready for use. The participants proceeded through each step while referring to the enclosed instructions. The 12 steps of the task are shown in Table 5.

**Table 5.** Details of the Google Nest Mini (product) task.

| Steps | Directions |
|---|---|
| 1st | Browse nest mini on Google Store. |
| 2nd | Open the box, take out the smart speaker. |
| 3rd | Read the instructions, turn on the smart speaker. |
| 4th | Install the Google app on your smartphone, select an account. |
| 5th | Connect apps and smart speakers using Wi-Fi connection with smartphone location information and router. |
| 6th | Open a Wi-Fi connection between the smart speaker and router using the app. |
| 7th | Follow the instructions in the app and using voice recognition on the smart speaker. |
| 8th | Connect and set various setting services in the app. |
| 9th | Play music on a smart speaker that has been set up. |
| 10th | Set alarm timers with smart speakers. |
| 11th | Listen to weather forecasts with smart speakers. |
| 12th | Evaluate your final user satisfaction with the Google Nest Mini. |

Eleven satisfaction datapoints and one final satisfaction datapoint were obtained for each participant, as shown in Figure 7. The resulting dataset for building the model consisted of a 12 × 25 matrix (12 features, 25 participants).

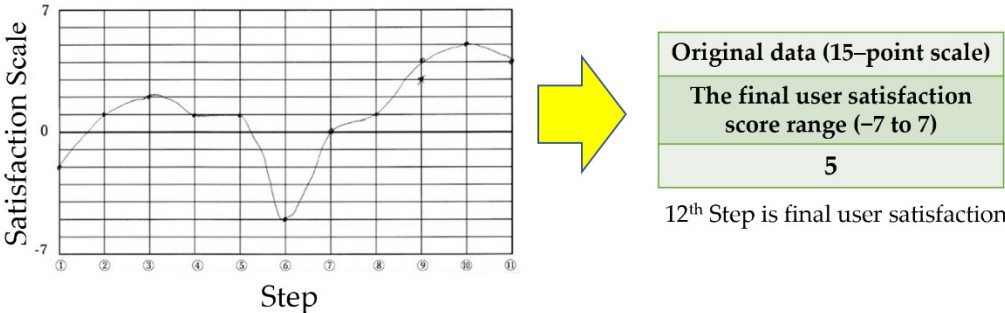

Steps 1–11 are steps/instructions.

**Figure 7.** Example of a customized UX curve with participant's final satisfaction.

The original dataset from our preliminary experiment showed an accuracy score less than 0.50, indicating that sufficiently accurate results were not obtained using the 15-point scale. Thus, we scaled down, converting the original dataset into two datasets, including a seven-point scale and a five-point scale (see Table 6). Dataset I comprised seven classes (from −3 to 3), and Dataset II comprised five classes (from −2 to 2). The actual results after shrinking showed three classes in Dataset I, and two classes in Dataset II due to zero samples in some classes.

**Table 6.** Original data scale was reduced by scaling down to improve the predictive performance.

| Meaning of Satisfaction Rating | Dataset I | | Meaning of Satisfaction Rating | Dataset II | |
|---|---|---|---|---|---|
| | Original Data | After Shrinking | | Original Data | After Shrinking |
| Extremely satisfied | 7 6 | 3 | Extremely satisfied | 7 6 | 2 |
| Satisfied | 5 4 | 2 | | 5 4 | |
| Slightly satisfied | 3 2 1 | 1 | Satisfied | 3 2 1 | 1 |
| Neutral | 0 | 0 | Neutral | 0 | 0 |
| Slightly unsatisfied | −1 −2 −3 | −1 | Unsatisfied | −1 −2 −3 | −1 |
| Unsatisfied | −4 −5 | −2 | | −4 | |
| Extremely unsatisfied | −6 −7 | −3 | Extremely unsatisfied | −5 −6 −7 | −2 |

Furthermore, the number of samples per class was found to increase when the number of classes decreased. The increased number of samples per class can be useful for building machine learning models; this is one advantage of shrinking.

Before processing the dataset, we used VIF to check multicollinearity for predictor variables (11 answers about satisfaction from the 11-item questionnaire), where the dependent variable was final user satisfaction. VIF values exceeding 5 or 10 indicate problematic collinearity [57]. All our VIF values were under 5.

In the current study, the first stage of the experiment to predict final user satisfaction using momentary UX was through the satisfaction survey form. For website evaluation, we used a satisfaction survey form at the bottom of the webpage to be filled out after the completion of each task. For product evaluation, we requested that the satisfaction survey

be manually evaluated during product set up. We found that it may not be easy to gather these satisfaction scores in an actual product evaluation situation. Future research could use other techniques for product evaluation to collect momentary UX data, such as facial expression or gaze data.

### 3.3. Evaluation

Several studies have attempted to demonstrate that SVM and KNN algorithms can perform well with small datasets [58,59]. Thus, we selected seven appropriate machine learning algorithms: SVM [36] including SVM with linear kernel, SVM with sigmoid kernel, SVM with RBF kernel, SVM with polynomial kernel, logistic regression [40], K-nearest neighbors (KNN) [39], and multilayer perceptron (MLP) [42]. Each model was trained by these algorithms using the datasets, and then classification models were tuned with various hyperparameters while evaluating machine learning models by a random search method to provide the best performance [60].

In Experiments I and II (Travel agency website and Google Nest Mini, respectively), we found an unequal distribution of classes within the datasets; for example, the ratios of seven classes in Experiment I (class "−3", class "−2", class "−1", class "0", class "1", class "2", and class "3") were 1, 0, 4, 1, 13, 25, and 6, respectively. As indicated in Section 2.4, multiple techniques exist for dealing with imbalanced sample distributions. Oversampling the minority class is one such approach used in data science [46]. This can be achieved by synthesizing new examples from the minority class in the training dataset prior to fitting a model. This can balance the class distribution and be highly effective for the created model. For example, in Experiment I, the number of samples increased from 50 to approximately 150. In Experiment II, the number of samples increased from 25 to approximately 36. By checking that the number of minority and majority class samples were equal, we confirmed that the imbalance disappeared. The most basic method involves creating examples from the minority class; even though these examples add no new information to the model, they can be created by combining existing data. Thus, we selected the synthetic minority oversampling technique, or SMOTE [46], based on results of the preliminary experiment.

One issue is that oversampling before performing cross-validation allows leakage from the test data into the training data. Because of the overlap between training and test data, this can lead to an optimistic bias in performance evaluation, as shown in Figure 8. This is why we used SMOTE oversampling techniques inside the cross-validation (CV) loop in the evaluation step. Oversampling inside the CV loop [61] is appropriate for revealing the model's true performance.

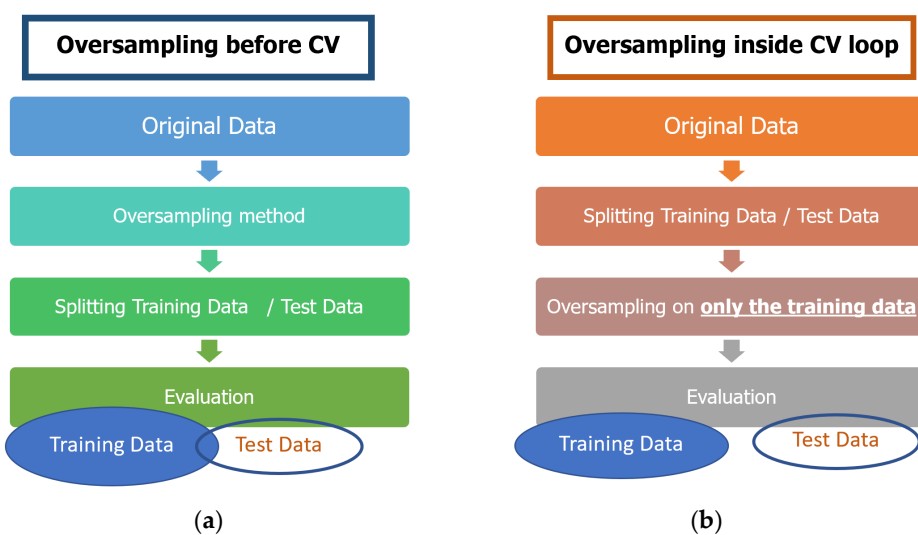

**Figure 8.** Comparison between (**a**) oversampling before the cross-validation loop and (**b**) oversampling inside the cross-validation loop.

In the evaluation step, two conventional methods were used to evaluate the performance of each classification model as follows: (1) leave-one-out cross-validation (LOOCV) [62], as shown in Figure 9 and (2) validation with training (80%)/test (20%) splitting by three indices: accuracy, recall, and precision, as shown in Figure 10 [63].

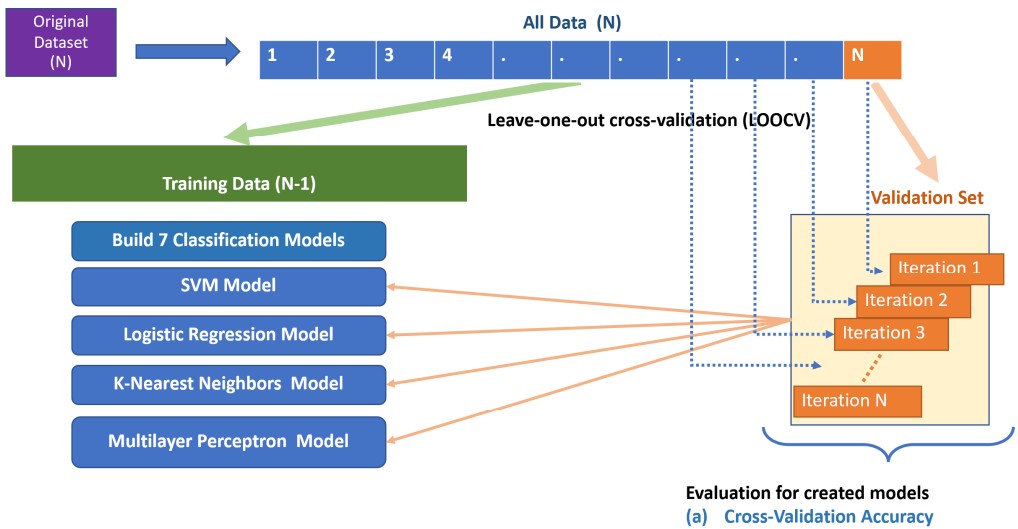

**Figure 9.** Evaluation workflow for created models using leave-one-out cross-validation (LOOCV).

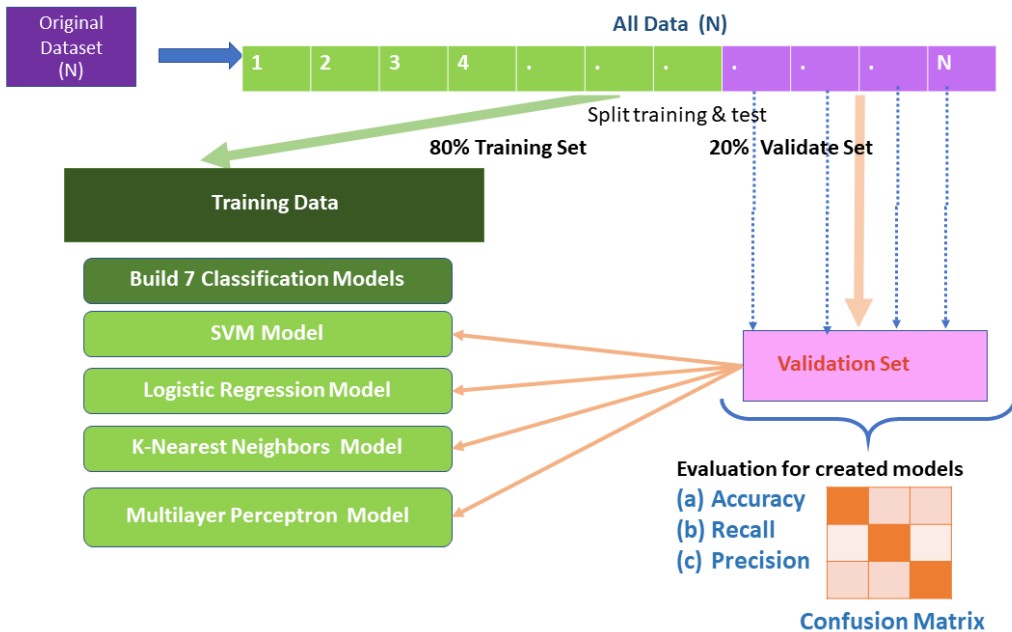

**Figure 10.** Evaluation workflow for created models using data splitting.

Moreover, performance can be measured using the percentage of accuracy observed in both data sets to conclude on the presence of overfitting. Overfitting is characterized by high accuracy for a classifier when evaluated on the training set but low accuracy when evaluated on a separate test set [64]. In our experiments, we confirmed that the accuracy score of the testing set was nearest when compared with that of the training set. Thus, all models were not overfitting.

In data science, data splitting according to an 80/20 ratio between the training set and test set provides the most practice for the machine learning model. The performance of each classification model is reported in terms of accuracy, precision, and recall. Accuracy is the most intuitive performance measure; namely, the ratio of correctly predicted observation to total observations. Precision is the ratio of correctly predicted positive observations to total predicted positive observations. Recall is the ratio of correctly predicted positive observations to all observations in the actual class.

## 4. Results

### 4.1. Results from Experiment I: Service Usage with Travel Agency Site

In Experiment I (travel agency website), we measured the combination of oversampling techniques with the created machine learning models, including SVM with polynomial kernel, SVM with RBF kernel, SVM with linear kernel, SVM with sigmoid kernel, K-nearest neighbors, logistic regression, and multilayer perceptron techniques. We then compared the performance between polynomial kernel SVM and polynomial kernel SVM with oversampling.

Table 7 shows a comparison of the classification models' performance for a combined synthetic minority oversampling technique (SMOTE) and machine learning techniques for two datasets (seven classes and five classes). Performance was measured by LOOCV and splitting the data into two subsets: training and test. The accuracy score can range between 0.00 and 1.00; a higher value indicates higher accuracy.

**Table 7.** Performance scores of created models from travel agency website.

| Scores | | Dataset | SVM Poly | SVM RBF | SVM Linear | SVM Sigmoid | KNN | LR | MLP |
|--------|--|---------|----------|---------|------------|-------------|-----|----|-----|
| LOOCV | Cross-Validation Accuracy | I (7 Classes) | 0.93 | 0.79 | 0.80 | 0.50 | 0.84 | 0.72 | 0.80 |
| | | II (5 Classes) | 0.90 | 0.87 | 0.88 | 0.45 | 0.80 | 0.87 | 0.84 |
| Split for training/test (80/20) | Accuracy | I (7 Classes) | 0.87 | 0.60 | 0.73 | 0.33 | 0.73 | 0.60 | 0.67 |
| | | II (5 Classes) | 0.93 | 0.93 | 0.86 | 0.54 | 0.86 | 0.89 | 0.93 |
| | Recall | I (7 Classes) | 0.87 | 0.60 | 0.73 | 0.33 | 0.73 | 0.60 | 0.67 |
| | | II (5 Classes) | 0.93 | 0.93 | 0.86 | 0.54 | 0.86 | 0.89 | 0.93 |
| | Precision | I (7 Classes) | 0.90 | 0.64 | 0.85 | 0.21 | 0.75 | 0.70 | 0.65 |
| | | II (5 Classes) | 0.96 | 0.95 | 0.87 | 0.42 | 0.88 | 0.90 | 0.93 |

SVM = support vector machine; Poly = polynomial kernel; LR = logistic regression; KNN = K-nearest neighbors; MLP = multilayer perceptron.

SVM with polynomial kernel using Dataset I (seven classes) had the highest LOOCV accuracy score (0.93). Moreover, each model was evaluated by splitting the training and test set techniques. SVM with polynomial kernel using Dataset I provided the highest accuracy (0.87), recall (0.87), and precision (0.90) scores.

However, SVM with the polynomial kernel using Dataset II (five classes) also showed the highest LOOCV (0.90). It also provided the highest scores for accuracy (0.93), recall (0.93), and precision (0.96).

Based on the results shown in Table 7, we then focused on SVM with the polynomial kernel. Table 8 summarizes the results of comparisons between polynomial kernel SVM and polynomial kernel SVM with oversampling into the cross-validation step. Polynomial kernel SVM with oversampling into the cross-validation step provided the highest cross-validation accuracies (0.93 and 0.90) on Datasets I and II, respectively. Moreover, the accuracy of polynomial kernel SVM with oversampling into the cross-validation step was higher than for polynomial kernel SVM without oversampling.

**Table 8.** Comparison of performance between polynomial kernel SVM and polynomial kernel SVM with oversampling into the cross-validation step (travel agency website).

| Model Performance | | Dataset I: 7 Classes (7-Point Scale Data) | | Dataset II: 5 Classes (5-Point Scale Data) | |
|---|---|---|---|---|---|
| | Score | Polynomial Kernel SVM | Polynomial Kernel SVM with Oversampling into the Cross-Validation Step | Polynomial Kernel SVM | Polynomial Kernel SVM with Oversampling into the Cross-Validation Step |
| LOOCV | Cross-Validation Accuracy | 0.48 | 0.93 | 0.72 | 0.90 |
| Split for training/test (80/20) | Accuracy | 0.40 | 0.87 | 0.70 | 0.93 |
| | Recall | 0.40 | 0.87 | 0.70 | 0.93 |
| | Precision | 0.65 | 0.90 | 0.61 | 0.96 |

Comparing classification results from two datasets differing in the number of classes revealed differences in accuracy scores between Datasets I (seven classes) and II (five classes). Overall, the accuracy with Dataset II was better than that with Dataset I, which suggests that the accuracy depends on the number of classes.

### 4.2. Results from Experiment II: Product Usage with Google Nest Mini

In Experiment II (use of Google Nest Mini), the models were validated by LOOCV. The results show that the SVM with polynomial kernel model provided the highest accuracy (Table 9). The correct answer rate when using the SVM with polynomial kernel method was the highest, at 0.76, suggesting the high effectiveness of this proposed method. Moreover, comparison of the classification result from two datasets differing in the number of classes revealed differences in accuracy scores between Datasets I (seven classes) and II (five classes). Furthermore, the accuracy score with Dataset II was higher than that with Dataset I. Taken together, these results confirm that the proposed framework is feasible, and it is possible to predict final user satisfaction guided by momentary UX data to answer product-satisfaction-related questions.

**Table 9.** Performance scores of created models from Google Nest Mini usage.

| Scores | | Dataset | SVM Poly | SVM RBF | SVM Linear | SVM Sigmoid | KNN | LR | MLP |
|---|---|---|---|---|---|---|---|---|---|
| LOOCV | Cross-Validation Accuracy | I (7 Classes) | 0.60 | 0.52 | 0.52 | 0.16 | 0.52 | 0.44 | 0.40 |
| | | II (5 Classes) | 0.76 | 0.68 | 0.64 | 0.32 | 0.68 | 0.68 | 0.48 |
| Split for training/test (80/20) | Accuracy | I (7 Classes) | 0.88 | 0.80 | 0.80 | 0.20 | 0.40 | 0.20 | 0.60 |
| | | II (5 Classes) | 0.86 | 0.60 | 0.80 | 0.40 | 0.40 | 0.40 | 0.60 |
| | Recall | I (7 Classes) | 0.88 | 0.80 | 0.80 | 0.20 | 0.40 | 0.20 | 0.60 |
| | | II (5 Classes) | 0.86 | 0.60 | 0.80 | 0.40 | 0.40 | 0.40 | 0.60 |
| | Precision | I (7 Classes) | 0.92 | 0.87 | 0.80 | 0.60 | 0.37 | 0.20 | 0.67 |
| | | II (5 Classes) | 0.89 | 0.60 | 0.85 | 0.53 | 0.53 | 0.53 | 0.87 |

SVM = support vector machine; Poly = polynomial kernel; LR = logistic regression; KNN = K-nearest neighbors; MLP = multilayer perceptron.

Based on the results shown in Table 9, we focused on SVM with the polynomial kernel. Table 10 summarizes the results of comparisons between polynomial kernel SVM and polynomial kernel SVM with oversampling into the cross-validation step. Polynomial kernel SVM with oversampling into the cross-validation step provided the highest cross-validation accuracies (0.60 and 0.76) with Datasets I and II, respectively. Moreover, the accuracy of polynomial kernel SVM with oversampling into the cross-validation step was higher than that of polynomial kernel SVM without oversampling.

**Table 10.** Comparison of performance between polynomial kernel SVM and polynomial kernel SVM with oversampling into the cross-validation step (Google Nest Mini usage).

| Model Performance | | Dataset I: 7 Classes (7-Point Scale Data) | | Dataset II: 5 Classes (5-Point Scale Data) | |
|---|---|---|---|---|---|
| **Score** | | **Polynomial Kernel SVM** | **Polynomial Kernel SVM with Oversampling into the Cross-Validation Step** | **Polynomial Kernel SVM** | **Polynomial Kernel SVM with Oversampling into the Cross-Validation Step** |
| LOOCV | Cross-Validation Accuracy | 0.52 | 0.60 | 0.60 | 0.76 |
| Split for training/test (80/20) | Accuracy | 0.60 | 0.88 | 0.60 | 0.86 |
| | Recall | 0.60 | 0.88 | 0.60 | 0.86 |
| | Precision | 0.50 | 0.92 | 0.80 | 0.89 |

When comparing the classification results from two datasets differing in the number of classes, differences in cross-validation accuracy between Datasets I (seven classes) and II (five classes) emerged. Overall, the cross-validation accuracy with Dataset II was better than with Dataset I, which suggests that accuracy depends on the number of classes.

## 5. Discussion

The present study was designed to predict final user satisfaction by machine learning techniques based on momentary UX Curve data. This study has several research implications, as discussed below.

### 5.1. Experiment I: Service Usage with Travel Agency Site

In the evaluation of service usage, performed with a travel agency website, we found that the SVM with the polynomial kernel algorithm provided the highest cross-validation accuracy, at 0.93; all other algorithms scored lower, with the next-highest being KNN, at 0.84, and slightly higher than the rest. Thus, SVM and KNN appear to be good at predicting final user satisfaction. To test the performance of these machine learning methods, we considered recall and precision on 20% testing and 80% training data. For travel agency website usage, SVM with polynomial kernel with both five and seven classes yielded the highest recall (0.87) and precision (0.90) among the seven candidate algorithms. Several previous studies have reported that recall and precision with imbalanced datasets may be poor [65] and lead to an optimistic bias in performance validation even after oversampling datasets [61]. In this study, however, recall and precision with oversampling resulted in data that were better than the original data, as shown in Tables 8 and 10. Two possible explanations for these improved results are, first, that we optimized the machine learning model by finding the best parameters for the dataset, and second, we performed oversampling during the cross-validation loop, which is the correct way to handle imbalanced data. Hence, to reveal the true performance of the model, it is appropriate that oversampling be conducted inside the cross-validation loop [61].

It is conceivable that the dimension of a dataset might be one factor influencing predictive performance. Some authors have reported that SVM and KNN might perform well on small datasets [58,59]. Moreover, in this study, accuracy was consistently higher with five classes (Dataset II) than that with seven classes (Dataset I), which suggests that accuracy depends on the number of classes. However, the ability of SVM and KNN to predict final user satisfaction should be further examined using other kinds of services.

### 5.2. Experiment II: Product Usage with Google Nest Mini

To evaluate product usage, we used a Google Nest Mini task and found that SVM with polynomial kernel with five classes had the highest cross-validation accuracy, at 0.76. Moreover, SVM with polynomial kernel with five classes had the highest recall (0.86) and

precision (0.89) of the seven candidate algorithms. Again, SVM with polynomial kernel performs better when the dataset has few classes.

Comparing oversampling and no oversampling revealed that the former five classes resulted in high cross-validation accuracy, at more than 76%. In this context, it is noteworthy that the use of momentary UX during service usage and classification models had the highest predictive accuracy for final user satisfaction.

*5.3. Findings*

With regard to assessment of the use of momentary UX and classification models, the most interesting finding was that momentary UX and machine learning can predict final user satisfaction, which is important for users' decisions about further use or whether they recommend products or services to other people. One unanticipated finding was that polynomial kernel SVM with an oversampling technique achieved the best classification accuracy (more than 90%). These results match those of machine learning studies where polynomial kernel SVM also performed better with oversampling because a higher degree of polynomial kernel, which is one of the parameters of the SVM algorithm, allows a more flexible decision boundary [66].

In this investigation, the aim was to predict final user satisfaction using momentary UX data and machine learning techniques. The results show that the machine learning process can help in predicting final user satisfaction in at least two contexts: Experiment I, service usage, and Experiment II, product usage.

The strongest feature of our proposed method is that it is based on data supporting the idea of the relationship between UX time intervals: momentary UX might affect episodic UX (final user satisfaction). Due to practical constraints, our preliminary study did not extend to evaluations involving a wider variety of products or services, and so we are cautious about extrapolating to other situations. Nevertheless, the study has demonstrated significant relationships between momentary UX data and final user satisfaction, which is consistent with the argument of Feng and Wei (2019) that a first-time user experience is generally seen as a factor related to long-term user experience [16].

## 6. Conclusions

Customer relationship management is a tool to improve both the business and customer satisfaction with products or service [19]. It is generally accepted that CRM provides essential feedback and reflective data from the customers' perspective, including their opinions, preferences, and past UX regarding to products or services. These data and information are used to improve communication with customers to enhance value and satisfaction.

In this study we aimed to predict final user satisfaction using momentary UX data and machine learning techniques. The findings indicate that machine learning techniques such as polynomial kernel SVM can comprehend participants' momentary UX and make better predictions than six other machine learning algorithms concerning their final user satisfaction. Moreover, machine learning integrated with the oversampling technique yielded higher accuracy than that without oversampling. This technique integrated with the oversampling method could deal with imbalanced classes by synthesizing new samples and adjusting the class distribution of a data set.

The study was divided into two different experiments, the first concerning evaluation of a service (travel agency website), and the second concerning a product (Google Nest Mini). For service usage with the travel agency site, the results showed that SVM with polynomial kernel had the highest cross-validation accuracy, at 0.93. For product usage with Google Nest Mini, the results showed that SVM with polynomial kernel again had the highest cross-validation accuracy, at 0.76. The proposed method, therefore, shows promise for accurately predicting final user satisfaction using machine learning techniques; it facilitates classification and estimation of final user satisfaction based on momentary UX Curve data.

### 6.1. Theoretical Implications

Our study has contributed to knowledge in the field in various ways.

First, our contribution relates to the outcomes of UX. Data of time sequence questionnaire or UX curve is often difficult to understand and analyze. We found the relationship between momentary UX and episodic or cumulative UX, which is related to final user satisfaction. Our study indicates how understanding of momentary UX data can help determine the final user satisfaction during the changes of UX curve [32].

Second, machine learning like SVM could accurately predict final user satisfaction and contribute towards developing products and services by analyzing the UX obtained from CRM [19]. Hence, we need to monitor the momentary UX carefully.

Third, combining and integrating machine learning and oversampling techniques could constitute a new approach for improving the predictive accuracy of final user satisfaction.

This finding shows the relevance of considering UX in the analysis of customer satisfaction.

### 6.2. Practical Implications

The majority of businesses that consider adopting a CRM system are looking for a way to improve the quality and consistency of their relationships with customers and build customer loyalty. UX data from CRM has gradually become the main source of businesses' sustainable competitive advantage. In terms of the practical implication of this study, the result of our proposed method is that it is based on data supporting the idea of the relationship between UX time intervals: momentary UX might affect episodic UX (final user satisfaction). The understanding of those aforementioned relationships could provide the best UX for customers, build a good brand image, and launch customer-centric marketing campaigns. It can help businesses to achieve a better user satisfaction and the goals of sustaining long-term competitive advantages. For example, in service industry, the product or service developers could discover the worst points of products or services at which a customer requires assistance during product or service usage. They need to ensure that customers can finish their transaction without difficulty in different usage situations. As a result, the understanding of momentary UX could boost overall customer satisfaction as well as repeat purchase rate, maintain long-term sustainable customer satisfaction and achieve sustainability. It could help to understand customers better and thus enhance communication with stakeholders with regard to efficiency, performance, and sustainability of products or services.

### 6.3. Limitations and Future Research

Our study confirmed the relationships between momentary UX data and final user satisfaction from evaluations involving products and services. In both experiments, the participants were university students. However, the low number of samples is one limitation of this study. Further validation of the methods requires studies with larger sample sizes. Regarding future work, it may also be possible to introduce other measures and features such as eye movement data and operation time data, with the aim of improving upon the predictive performance reported here.

**Author Contributions:** Conceptualization, K.K. and N.N.; methodology, K.K. and N.N.; software, K.K.; validation, K.K.; formal analysis, K.K.; investigation, K.K.; resources, K.K. and N.N.; data curation, K.K.; writing—original draft preparation, K.K.; writing—review and editing, K.K. and N.N. All authors have read and agreed to the published version of the manuscript.

**Funding:** This research was funded by JSPS KAKENHI, grant number JP20K12511.

**Institutional Review Board Statement:** Not applicable.

**Informed Consent Statement:** Informed consent was obtained from all subjects involved in the study.

**Data Availability Statement:** Not applicable.

**Acknowledgments:** The authors also thank Misaki Imamura and Ayato Kakegawa for supporting experiments.

**Conflicts of Interest:** The authors declare no conflict of interest.

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
