# Peer review of "Predicting Final User Satisfaction Using Momentary UX Data and Machine Learning Techniques"

_jtaer, doi:10.3390/jtaer16070171_

Round 1

Reviewer 1 Report

  • “The participants were 25 and 50 university students who were asked to evaluate a product (Experiment I) or service (Experiment II), respectively.” Firstly, 25 or 50 students are very limited to develop the machine learning models. Although the authors said using oversampling to handle imbalanced data, the original data set is still very limited especially the number of classes of models are five and seven. Since the data is too limited, the modeling results maybe not be acceptable.
  • “Oversampling the minority class is one such approach used in data science. “ The details were missed in the manuscript.
  • How could authors make sure there is no overfitting in your models? The authors used leave-one-out cross-validation (LOOCV) or splitting for training/test (80/20) for their models. For new data, how could the authors make sure it is not overfitting?
  • The experiment subject were university students. The experiment is limited since the experiment subject should be from different groups and different conditions. The authors did not explain why they only choose students as experiment subjects.
  • “The results of this investigation show that predictive performance for service usage (assessed using the travel agency site) was better than for product usage (assessed using Google Nest Mini). “ Does these 25 students of the experiment I covered in 50 students of experiment II? Why not using the same number of experiment subjects for the experiment I and experiment II? Do the authors consider using the same subjects for the experiment I and experiment II? Currently, the results of experiment I and experiment II are not comparable and maybe not be acceptable.
  • The details of developing these models are missing. A simple introduction to these seven models is still needed. Did the authors tuning the models or using random hyper-parameters?
  • The literature on using machine learning models to predict user experience or similar target is limited.
  • The writing of the paper is well.

Reviewer 2 Report

Customer satisfaction analysis is an important and current problem. The selection of the appropriate method to analyze customer satisfaction guarantees the correct functioning of the company's information and decision-making system. In this context, I consider the research presented in the article to be valuable for science and business practice. At the same time, I point out the following weaknesses of the article that should be removed.

In Figure 2, the authors present the Evaluation of final user satisfaction. How do the authors understand the term "final user satisfaction" itself? Is this a synonym for "cumulative user satisfaction" or “episodic UX”? I have the impression that the authors use these terms interchangeably in the article.

It is not entirely clear to me whether the authors understand the term “momentary  UX” (defined in the purpose of the article) as “usage time” (Figure 1), or rather the method?

Why was the research sample limited to students aged 21-24 only?

How were students recruited for the study? Was it a purposive sampling? Why was the 15-point scale used at the beginning of the study? Why was the second study used a 21-point scale first? 

Round 2

Reviewer 1 Report

  1. For example, in Experiment I, the number of samples increased from 50 to approximately 150, until imbalance disappeared. In Experiment II, the number of samples increased from 25 to approximately 36, until imbalance disappeared. “ How do the authors make sure the imbalance disappear? This is not clear.
  2. Generally, the reference number should be at the end of the sentence.
  3. Please add references to machine learning models in table 2. In general, all definitions that are not your contribution need the proper reference.
  4. Please correct the writing “Many studies have tried to estimate the satisfaction of users using various machine learning techniques.[11–13] Matsuda[11] studied the satisfaction level of tourists during sightseeing by using the tourists’ unconscious, natural actions.”
  5. In section 6 conclusion, “Moreover, machine learning integrated with the oversampling technique yielded higher accuracy than the original data.” Did the authors generate machine learning results using original data to compare with oversampling and without oversampling? If not, please correct.
  6. The low number of samples is the limitation of this paper, please include this in the conclusion.
